# Do Metadata and Appearance of the Retrieved Webpages Affect LLM's Reasoning in Retrieval-Augmented Generation?

**Cheng-Han Chiang**
National Taiwan University,
Taiwan
dcml0714@gmail.com

**Hung-yi Lee**
National Taiwan University,
Taiwan
hungyilee@ntu.edu.tw

## Abstract

Large language models (LLMs) answering questions with retrieval-augmented generation (RAG) can face conflicting evidence in the retrieved documents. While prior works study how textual features like perplexity and readability influence the persuasiveness of evidence, humans consider more than textual content when evaluating conflicting information on the web. In this paper, we focus on the following question: When two webpages contain conflicting information to answer a question, does non-textual information affect the LLM's reasoning and answer? We consider three types of non-textual information: (1) the webpage's publication time, (2) the source where the webpage is from, and (3) the appearance of the webpage. We give the LLM a Yes/No question and two conflicting webpages that support yes and no, respectively. We exchange the non-textual information in the two webpages to see if the LLMs tend to use the information from a newer, more reliable, and more visually appealing webpage. We find that changing the publication time of the webpage can change the answer for most LLMs, but changing the webpage's source merely changes the LLM's answer. We also reveal that the webpage's appearance has a strong causal effect on Claude-3's answers. The codes and datasets used in the paper are available at https://github.com/d223302/rag-metadata.

## 1 Introduction

Retrieval-augmented LLMs (Guu et al., 2020; Lewis et al., 2020) respond to user queries by considering the documents retrieved from external knowledge sources, ranging from Wikipedia (Chen et al., 2017) to the whole Web (Piktus et al., 2021; Nakano et al., 2021). As the knowledge source scales up and the user queries become more diverse, the retrieved contents can contain conflicting information. Extensive prior works have explored how

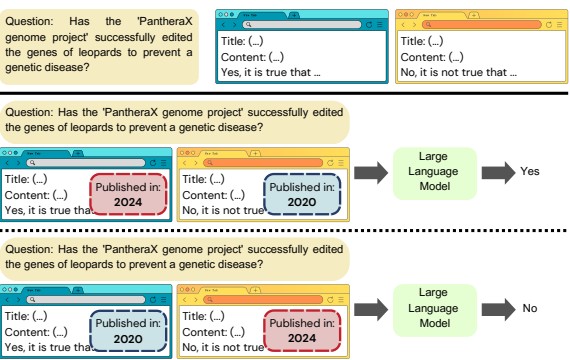

Figure 1: Given a Yes/No question and two documents that support Yes and No, respectively, we add a type of non-textual information (publication date in this figure) to both documents with different values. We swap the non-textual information in the two documents and see whether the LLM's answer to the question is different.

LLMs reason over conflicting documents (Chen et al., 2022; Jin et al., 2024; Xu et al., 2024).

When humans are presented with contradicting evidence that leads to different answers, we use multiple strategies to reason over the searched webpages (Wathen and Burkell, 2002; Metzger et al., 2010; Kąkol et al., 2013; Kakol et al., 2017), including the credibility of the sources (Tandoc Jr, 2019; Bates et al., 2006) and the arguments in the documents (Fogg et al., 2003). Then, what about LLMs? What evidence do LLMs find convincing when conflicting information exists in the retrieved documents? To understand this, Wan et al. (2024) constructs CONFLICTINGQA, consisting Yes/No questions and documents extracted from real webpages that support both stances. They analyze what **text features** in the document make the LLM more inclined to agree with the stance in the document.

While Wan et al. (2024) provide valuable insights into how text features affect a webpage's credibility for LLMs, they do not explore how the information *beyond* the document's content affects the LLM's decision. This is because most retrieval-

augmented LLMs only take the titles and the textual contents as input while discarding all the metadata of the webpage, including URL and webpage publication times (Karpukhin et al., 2020; Gao et al., 2023). Consequently, it is unclear whether LLMs can use metadata of the webpages for reasoning when these metadata are provided. Additionally, given the wide application of using vision language models (VLM) (Zhu et al., 2023; Chen et al., 2023; Wang et al., 2024) for web navigation and webpage question answering (Koh et al., 2024; Liu et al., 2024b; Cheng et al., 2024), it is unclear how the visual appearance of the webpage affects how VLMs reason based on the webpage.

In this paper, we explore the following research question: *Can the metadata and appearance of the retrieved webpages affect the LLM's answer?* In our paper, we will use the term **non-textual information** to refer to the information in a webpage other than its title and textual contents, which can include the webpage's metadata and its appearance.[1] Inspired by how a human's reasoning can be affected by non-textual information, including (**1**) the webpage's publication time (Sundar et al., 2007; Westerman et al., 2014), (**2**) the source's credibility (Bates et al., 2006; Tandoc Jr, 2019), and (**3**) the appearance of the webpage (Fogg et al., 2003), we want to know if these factors can affect LLM's answer. We will use the terms *document* and *webpage* interchangeably as we simulate the case when the documents are webpages retrieved from the Web.

We give an LLM a Yes/No question and two documents supporting contradicting stances, with non-textual information incorporated in the documents. We exchange the non-textual information in the two documents to see if the LLM's answers change and whether the LLM's answer agrees with the stance of a webpage published more recently, from a more reliable source, or looks better. We conduct causal analyses to understand whether the non-textual information affects the LLM's answer. Additionally, we check the LLM's responses to see if it mentions non-textual information.

To the best of our knowledge, we are the first to explore the role of non-textual information in RAG with conflicting evidence. We have the following intriguing observations:

---

[1] While the metadata of a webpage are still presented in texts, we use the term *non-textual information* to refer to webpage metadata and appearance for the sake of simplicity.

- Most LLMs agree with the stance of a webpage published more recently.

- Although some LLMs mention where the document is from, they do not align their answers with the stances of more reliable sources.

- All Claude-3 models (Anthropic, 2024) tend to adopt the answer from a CSS-formated webpage compared with a plain HTML webpage.

## 2 Experiment Setup

To answer whether LLM's answer and reasoning can be affected by the non-textual information, we give the LLM a Yes/No question and a pair of documents that support Yes and No, respectively. The two documents include their respective non-textual information. We observe whether the LLM's answer can be changed by exchanging only the non-textual information in the two documents and whether the LLM's reasoning mentions the non-textual information. The overall experiment setup is shown in Figure 1.

### 2.1 Dataset

We use CONFLICTINGQA created by Wan et al. (2024) and CONFLICTINGQA-FAKE we create ourselves in our experiments.

#### 2.1.1 CONFLICTINGQA

CONFLICTINGQA is designed to simulate realistic scenarios where an LLM may encounter contradicting evidence in RAG. The questions in CONFLICTINGQA are controversial real-world Yes/No questions, and each question is paired with documents retrieved from the Web that support two stances (Yes or No). We preprocess CONFLICTINGQA and obtain 355 questions. We present detailed statistics and pre-processing steps in Appendix A.1.

#### 2.1.2 CONFLICTINGQA-FAKE

The questions in CONFLICTINGQA are based on real-world controversies, and LLMs may already have their own stances. While we ask the LLMs only to use the documents given to them to answer the question, it is unclear whether the LLMs rely on their own stance to answer the question.

To address the aforementioned issue, we collected 125 Yes/No questions generated by GPT-4 (OpenAI, 2023) about a non-existent entity. The questions are generated based on the 191 categories in Wan et al. (2024), detailed in Appendix A.2. An example question is shown in Figure 1, which is

about a fake scientific project called "PantheraX genome project". We include more examples in Table 5 in the Appendix. For each question, we prompt GPT-4 to produce a document that supports a given stance (Yes or No) and a title for the document. To verify that the document indeed supports the desired stance used to generate the document, we prompt GPT-4 with the question and the generated document to see if GPT-4's answer matches the desired stance. If GPT-4's answer does not match the desired stance, we discard the document. We elaborate on how we prompt GPT-4 in Appendix A.2.

After this process, we obtain 125 questions, each with two documents supporting two stances (Yes or No). We call the resulting dataset CONFLICTINGQA-FAKE as they are based on fake entities that do not exist. An author manually examines all 125 questions to ensure the entities in those questions are non-existent, and the questions are reasonable.

The main distinction between CONFLICTINGQA and CONFLICTINGQA-FAKE lies in that all questions in CONFLICTINGQA-FAKE are about non-existent entities and all the *supporting documents* are documents generated by GPT-4, which are unlikely to be used as the training data of the LLMs. Consequently, the LLMs are less likely to hold any pre-existing stances on these questions. This simulates scenarios where LLMs encounter contradictory information about unknown entities.

## 2.2 LLMs

Our experiments include 10 LLMs: Llama-2-chat (7B, 13B) (Touvron et al., 2023), Llama-3-Instruct (8B and 70B) (Meta, 2024), Tulu-v2-dpo-7B (Ivison et al., 2023; Rafailov et al., 2024), GPT-4-turbo (OpenAI, 2023), GPT-4o (OpenAI, 2024), and Claude-3 (haiku, sonnet, and opus) (Anthropic, 2024). We sometimes do not refer to the LLMs using full names as long as there is no ambiguity.

## 2.3 Querying LLMs for an Answer

We query the LLM with a question and two contradicting webpages. The LLM is asked to only use the provided webpages to answer the question. We use two types of query to make the LLM answer: (1) Direct answer and (2) chain-of-thought (CoT) (Wei et al., 2022; Kojima et al., 2022).

**Direct answer**  Following Wan et al. (2024), we prompt the LLM to "answer with only 'Yes' or 'No'" and say nothing else. A simplified prompt

```
The following are two websites:
Website 1:
Title: [title_1]
Text: [content_1]

Website 2:
Title: [title_2]
Text: [content_2]

Website 1 is from the URL: [URL_1].
Website 2 is from the URL: [URL_2].

Using these two sources, answer the following question: "[question]"
Use    only    the    information    in    the    above    text
to    answer    the    question. Answer with only "Yes" or
"No". Do not answer with anything other than  "Yes" or "No".
```

Table 1: The simplified prompt for querying the LLM to give a Yes/No answer directly with the website URL added (Section 4). The texts highlighted in green are only used when the LLM is prompted to answer directly.

is shown in Table 1. For open-source LLMs, we compare the condition probability of Yes and No given the input prompts. For proprietary LLMs, we check if the response contains Yes or No.

**Chain-of-thought (CoT)**  Instead of forcing the LLM to answer with Yes/No and say nothing else, we prompt the LLM to "give a concise answer with explanations." We would like to see whether the LLM's answer can change when they can think step-by-step. This also allows the LLM to acknowledge the conflicting sources in the provided context and say the answer is inconclusive, which may be a desired behavior when the LLM is provided with conflicting answers (Chen et al., 2022). After obtaining the response from the LLM, we prompt ChatGPT-3.5 (OpenAI, 2022) to extract the final answer using three options: Yes, No, and Inconclusive.

For each question and a pair of documents, we query the LLM twice by exchanging the position of the two documents to avoid potential position bias of the LLM (Wang et al., 2023). If the answers when swapping the documents' positions are inconsistent, the LLM's answer is considered as N/A. The LLMs answer can be (1) Yes, (2) No, (3) N/A for the *direct answer* setting. The *CoT* answer can be (1) Yes, (2) No, (3) Inconclusive, where the LLM always finds the answer is inconclusive when we swap the order of the documents, and (4) N/A.

## 2.4 Understanding the Effect of Non-Textual Information to LLM's Answer

Given a Yes/No question and two documents supporting contradicting stances, we add non-textual information into the two documents and see whether non-textual information affects the LLM's answer. In this paper, we refer to a document sup-

porting "yes" as *yes-document*, denoted as $d_\checkmark$, and the document supporting "no" as *no-document*, denoted by $d_\times$. By adding non-textual information, we want to simulate the case as if the document is from a webpage retrieved from the Web. Motivated by how humans consider a webpage's credibility, we consider the following three factors: the webpage's publication time, the source where the webpage is from (e.g., Wikipedia or CNN News), and the appearance of the webpage.

For a fixed type of non-textual information, we conduct the following experiment to understand if changing the non-textual information affects the LLM's answer. First, for a question $q$ and two contradicting documents, we add non-textual information to the documents, where the non-textual information of yes-document takes the value $v_1$ and that of no-document takes the value $v_2$. We use $(d_\checkmark : v_1; d_\times : v_2)$ to denote the document pair added with non-textual information after the above process. How the non-textual information is added depends on the type of non-textual information, which will be explained in the respective sections. We use the question and two documents $(d_\checkmark : v_1; d_\times : v_2)$ to query the LLM. We denote the LLM's answer as $Y_q(d_\checkmark : v_1; d_\times : v_2)$; $Y_q = 0$ if the LLM's response is no; otherwise, $Y_q = 1$[2].

Next, we exchange the non-textual information $v_1$ and $v_2$ in the two documents to form $(d_\checkmark : v_2; d_\times : v_1)$, where the yes-document's non-textual information is $v_2$ while that of the no-document is $v_1$. We use the same question $q$ and two documents $(d_\checkmark : v_2; d_\times : v_1)$ to query the LLM and obtain the LLMs answer: $Y_q(d_\checkmark : v_2; d_\times : v_1)$.

### 2.4.1 Evaluation Metrics

We use **flip ratio** and **No%** to evaluate whether the answer changes before and after swapping the non-textual information in the LLM's response. Since the questions in CONFLICTINGQA-FAKE are fictional, we do not use accuracy as an evaluation metric as there is no ground truth.

**Flip ratio** We report the proportion of questions in the dataset whose answer changes when we swap the non-textual information in the documents;

---

[2]Note that $Y_q = 1$ can include the cases when the LLM's answer is Yes, Inconclusive, and N/A. We consider $Y_q$ as a binary variable for ease of using the McNemar test. Additionally, our goal is to understand whether changing the non-textual information changes the model's output, consequently, as long as the LLM's answer is different from its original prediction after flipping the non-textual information, we attribute this change to the non-textual information.

we call this the flip ratio. Since the LLM's inputs when giving the answer $Y_q(d_\checkmark : v_1; d_\times : v_2)$ and $Y_q(d_\checkmark : v_2; d_\times : v_1)$ only differ in the non-textual information, if the above two answers disagree, this can only stem from the modification to non-textual information. Note that we consider N/A, where the LLM's answer is inconsistent when swapping the **position** of the two documents, as a type of answer and falls in the type of $Y_q = 1$.

**No%** We calculate the average number of questions that the LLM answers No under a specific configuration of the non-textual information, e.g., $(d_\checkmark : v_1; d_\times : v_2)$ or $(d_\checkmark : v_2; d_\times : v_1)$. We call this number the No%. If No% for $(d_\checkmark : v_1; d_\times : v_2)$ is higher than that of $(d_\checkmark : v_2; d_\times : v_1)$, this indicates that $v_2$ tends to make the LLM to agree with the stance in that document.

### 2.4.2 Causal Analysis

We conduct causal analyses to see if changing the non-textual information causes the LLM to change its answer. We first introduce some backgrounds in causal inference (Hernán and Robins, 2010). Causal inference aims to know whether a treatment $S$ has a causal effect on an outcome $Y$; specifically, whether the outcome when the treatment is set to $s_1$, denoted as $Y(s = s_1)$, differs from the outcome when the treatment is set to $s_2$, denoted as $Y(s = s_2)$. If $Y(s = s_1) \neq Y(s = s_2)$, we say treatment $S$ has a causal effect on the outcome $Y$.

Here, we consider $Y_q$, the LLM's answer for $q$, as the outcome. $Y_q = 0$ when LLM answers No and $Y_q = 1$ otherwise. The treatment we consider is how the non-textual information in the two documents is set, which can be $(d_\checkmark : v_1; d_\times : v_2)$ or $(d_\checkmark : v_2; d_\times : v_1)$. We can calculate the proportion of questions whose $Y_q(d_\checkmark : v_1; d_\times : v_2) = 0$ but $Y_q(d_\checkmark : v_2; d_\times : v_1) = 1$; we also calculate the proportion of questions whose $Y_q(d_\checkmark : v_1; d_\times : v_2) = 1$ but $Y_q(d_\checkmark : v_2; d_\times : v_1) = 0$. By comparing the two proportions, we can understand if changing $(d_\checkmark : v_1; d_\times : v_2)$ into $(d_\checkmark : v_2; d_\times : v_1)$ makes the LLM change the answer to No more often or not. Since our outcome is binary and each question undergoes a pair of treatments, we use McNemar's test (McNemar, 1947) to see whether the outcomes of the two treatments are significantly different.

It is worth noting that comparing the No% before and after we exchange the non-textual information is not equivalent to calculating the flip ratio under these two settings. It is easy to construct cases that have the same No% but have different flip ratios. It

is also important to note that a high flip ratio does not guarantee that a treatment $S$ has a causal effect on the outcome $Y$. This is because the flip ratio only considers the total counts of pairs that change from $Y_q = 0$ to $Y_q = 1$ or from $Y_q = 1$ to $Y_q = 0$, while in our paired causal analysis (McNemar's test), we further consider the difference between the number of pairs that change from $Y_q = 0$ to $Y_q = 1$ and from $Y_q = 1$ to $Y_q = 0$ after swapping the non-textual information.

# 3 The Webpage's Publication Time

First, we focus on the publication time of the webpage, which is an important webpage metadata. Since removing the metadata and extracting only the textual content is the first step to pre-process a webpage, metadata, including publication time, is seldom used as input to the LLM in RAG (Chen et al., 2017; Wan et al., 2024). While prior works on time-dependent question-answering benchmarks consider the publication time of a webpage (Zhang and Choi, 2021; Kasai et al., 2023; Zhang and Choi, 2023), they do not thoroughly study the effect of the publication time on the LLM's answer in a controlled and causal way as we do. Moreover, compared with SITUATEDQA (Zhang and Choi, 2021) and REALTIME QA (Kasai et al., 2023), which are based on real-world entities, using CONFLICTINGQA-FAKE allows us to reduce the possibility of LLM relying on its parametric knowledge instead of the retrieved evidence.

## 3.1 Adding Publication Times to Documents

To add publication time to a pair of documents, we add the following sentence to each document in the next line of its title: "Website publication time: [date]." To understand whether LLMs prefer to trust and rely on more up-to-date documents among the two documents, we set one of the document's publication time to 2024-04-01 and another to 2020-04-01. We select these days since 2024-04-01 is newer than the knowledge cut-off date of all LLMs we use, while all LLMs should be trained on data collected after 2020.

We compare the LLM's answer when the input documents are set to $(d_✓ : 20; d_✗ : 24)$, where the yes-document's publication date is set to 2020-04-01 and that of no-document is set to 2024-04-01, and $(d_✓ : 24; d_✗ : 20)$, where the yes-document's publication date is set to 2024-04-01 and that of no-document is set to 2020-04-01.

When inserting the publication times into the documents, it might be important to tell the LLM today's date (Kasai et al., 2023). We are also interested in understanding how important it is to tell the LLM what date it is today. We consider two settings: (1) *no today*: we do not tell the LLM what date it is today in our input prompt.[3] (2) *today*: we add "Today is 2024/04/30." in the input prompt when prompting the LLM.

## 3.2 Experiment Results

We show the results of CONFLICTINGQA-FAKE in Table 2 and the results of CONFLICTINGQA in Table 8 in the Appendix; the following observation is mostly consistent between the two datasets.

**The flip ratio for most LLMs is much larger than 0.** This observation holds no matter if LLMs are asked to answer directly or provide CoT reasoning. This shows that simply exchanging the publication dates of the two documents can make the LLM's prediction different.

**No% for some models do not differ when varying the publication time.** For Llama-2-7B and Llama-2-13B, their No% does not change significantly under $(d_✓ : 20; d_✗ : 24)$ and $(d_✓ : 24; d_✗ : 20)$ when prompted to directly answer. When they are prompted to reason using CoT but today's date is not given, we also do not see the No% to be too different when swapping the document publication dates; in this case, we find that these two models merely mention the publication dates in their CoT reasoning. This shows that the two models may not use document publication times when answering questions with conflicting evidence.

**Telling Haiku and Tulu what the date is today can make a difference.** We observe that when we do not say what date today is in the prompt, the No% gap between $(d_✓ : 20; d_✗ : 24)$ and $(d_✓ : 24; d_✗ : 20)$ for Haiku is only 1.8% when prompted to direct answer and 0.8% when answer by CoT. However, when we explicitly prompted with today's date, the No% difference when swapping the publication dates significantly increases to 17.6% for the direct answer setting and to 35.2% for the CoT setting. This shows that the LLMs can be affected by whether the current time is provided when the retrieved documents contain time information.

**GPT-4-turbo says No more often when the no-document is newer.** Regardless of whether we

---

| LLM | Direct Answer | | | | | | CoT | | | | | |
|---|---|---|---|---|---|---|---|---|---|---|---|---|
| | no-today | | | today | | | no-today | | | today | | |
| | No% | | Flip ratio | No% | | Flip ratio | No% | | Flip ratio | No% | | Flip ratio |
| | ✓:20 ✗:24 | ✓:24 ✗:20 | | ✓:20 ✗:24 | ✓:24 ✗:20 | | ✓:20 ✗:24 | ✓:24 ✗:20 | | ✓:20 ✗:24 | ✓:24 ✗:20 | |
| GPT-4-turbo | 76.0 | 42.4 | 47.2 | 92.8 | 20.0 | 77.6 | 20.0 | 4.8 | 49.6 | 28.0 | 2.4 | 68.0 |
| haiku | 96.8 | 98.4 | 3.2 | 100.0 | 82.4 | 17.6 | 40.0 | 39.2 | 57.6 | 59.2 | 24.0 | 68.0 |
| sonnet | 84.0 | 73.6 | 28.8 | 99.2 | 26.4 | 73.6 | 1.6 | 0.8 | 42.4 | 17.6 | 0.0 | 72.8 |
| Llama-2-7B | 0.0 | 0.0 | 8.0 | 0.0 | 0.0 | 19.2 | 72.8 | 71.2 | 35.2 | 76.0 | 64.8 | 35.2 |
| Llama-2-13B | 99.2 | 99.2 | 0.8 | 100.0 | 96.8 | 3.2 | 51.2 | 52.8 | 30.4 | 45.6 | 36.8 | 42.4 |
| tulu-7B | 48.0 | 44.8 | 50.4 | 55.2 | 43.2 | 57.6 | 23.2 | 24.8 | 60.0 | 31.2 | 18.4 | 62.4 |
| Llama-3-8B | 89.6 | 76.0 | 21.6 | 99.2 | 32.8 | 66.4 | 21.6 | 21.6 | 73.6 | 40.8 | 9.6 | 88.0 |
| Llama-3-70B | 96.8 | 84.8 | 14.4 | 99.2 | 54.4 | 44.8 | 36.0 | 23.2 | 60.0 | 68.0 | 15.2 | 76.8 |

Table 2: The No% and the flip ratio (columns in red) on CONFLICTINGQA-FAKE when changing the website's publication date. ✓:20,✗:24 corresponds to $(d_\checkmark : 20; d_\chi : 24)$; ✓:24,✗:20 corresponds to $(d_\checkmark : 24; d_\chi : 20)$. The blocks highlighted in blue represent the pairs when there is a significant difference ($p$-value $< 0.01$) between the model's answer between $(d_\checkmark : 20; d_\chi : 24)$ and $(d_\checkmark : 24; d_\chi : 20)$ based on McNemar's test.

tell GPT-4-turbo the date of today or whether it is asked to directly answer or answer with CoT, GPT-4-turbo's No% is always higher when the no-document is newer. Still, we observe that the flip ratio and the No% gap between $(d_\checkmark : 20; d_\chi : 24)$ and $(d_\checkmark : 24; d_\chi : 20)$ increase when we explicitly tell GPT-4-turbo what date is today.

**Models with higher No% when the no-document is newer frequently mention the date in their CoT responses.** When prompted to answer by CoT, models including GPT-4-turbo and Llama-3 models have a **No%** much higher when $(d_\checkmark : 20; d_\chi : 24)$ compared with $(d_\checkmark : 24; d_\chi : 20)$. We use regular expressions to extract whether the model responses mention the date 2024 or 2020, and we find that for the above models, they mention the date in at least 32.8% of the responses for Llama-3-8B and as high as 93.6% for GPT-4. By scrutinizing the responses from these models, we find that they often say "*based on the more up-to-date source...*". This shows that these models can use the publication time to reason over the question.

**Changing Webpage publication dates causes the model to change their answers in most settings.** In Table 2, we highlight the pairs of results when swapping the publication dates of the webpages causes the LLM's answers to be significantly different based on McNemar's test. For all models, when prompted to reason with CoT, as long as today's date is provided, the LLM's answer is significantly different before and after swapping the publication dates. By comparing the No% between $(d_\checkmark : 20; d_\chi : 24)$ and $(d_\checkmark : 24; d_\chi : 20)$, we can see that the LLMs prefer to answer No more often when the no-document is newer. Based on the

above results, we conclude that changing the publication times of the document does have a causal effect on the responses of some LLMs.

## 4 Source of the Webpage

Next, we explore the source of the webpage. We are specifically interested in the case when documents are from sources that differ in credibility. We use the following pair of webpage sources: Wikipedia and WordPress. Wikipedia is a trustworthy source with mostly verified information, while WordPress is mostly personal blogs and does not guarantee its information's correctness. Conflicting information from diverse sources is an important topic in fact-checking (Vlachos and Riedel, 2014; Augenstein et al., 2019; Gupta and Srikumar, 2021; Khan et al., 2022; Glockner et al., 2022). We differ from them by using counterfactual analysis, i.e., swapping the sources of the documents, to understand the role of the source to LLMs in RAG.

### 4.1 Adding Source to Documents

For each question, the LLM will be prompted twice by (1) setting the yes-document from Wikipedia and the no-document from WordPress and (2) setting the yes-document from WordPress and no-document from Wikipedia. We denote the above two settings as $(d_\checkmark : Wk; d_\chi : WP)$ and $(d_\checkmark : WP; d_\chi : Wk)$ respectively. While we only show a pair of sources in the main content, we repeat the experiment on another pair of sources, CNN News and NaturalNews, a trustworthy news source and a website known for fake news, respectively, and the result using this pair of sources is similar to the results of using Wikipedia and WordPress, which

| | Direct Answer | | | | | | CoT | | | | | |
| | URL | | | Name | | | URL | | | Name | | |
| | No% | | Flip ratio | No% | | Flip ratio | No% | | Flip ratio | No% | | Flip ratio |
| LLM | ✓:WP ✗:Wk | ✓:Wk ✗:WP | | ✓:WP ✗:Wk | ✓:Wk ✗:WP | | ✓:WP ✗:Wk | ✓:Wk ✗:WP | | ✓:WP ✗:Wk | ✓:Wk ✗:WP | |
|---|---|---|---|---|---|---|---|---|---|---|---|---|
| GPT-4-turbo | 83.2 | 74.4 | 20.8 | 80.0 | 78.4 | 19.2 | 10.4 | 13.6 | 31.2 | 12.0 | 9.6 | 34.4 |
| haiku | 98.4 | 98.4 | 2.4 | 99.2 | 97.6 | 2.4 | 47.2 | 43.2 | 55.2 | 40.0 | 37.6 | 53.6 |
| sonnet | 73.6 | 82.4 | 28.0 | 82.4 | 76.0 | 26.4 | 1.6 | 3.2 | 38.4 | 0.8 | 0.8 | 44.0 |
| Llama-2-7B | 0.0 | 0.0 | 3.2 | 0.0 | 0.0 | 1.6 | 72.8 | 71.2 | 34.4 | 25.6 | 28.8 | 60.0 |
| Llama-2-13B | 99.2 | 98.4 | 1.6 | 99.2 | 99.2 | 0.8 | 46.4 | 49.6 | 30.4 | 40.8 | 42.4 | 36.0 |
| tulu-7B | 54.4 | 46.4 | 44.8 | 36.8 | 33.6 | 55.2 | 19.2 | 23.2 | 63.2 | 30.4 | 28.0 | 45.6 |
| Llama-3-8B | 62.4 | 56.8 | 41.6 | 61.6 | 49.6 | 48.8 | 12.8 | 17.6 | 70.4 | 17.6 | 16.8 | 59.2 |
| Llama-3-70B | 94.4 | 89.6 | 10.4 | 91.2 | 92.0 | 8.8 | 26.4 | 30.4 | 66.4 | 22.4 | 32.8 | 60.8 |

Table 3: The No% and the flip ratio (columns in red) on CONFLICTINGQA-FAKE when changing the webpages' sources. ✓:WP,✗:Wk corresponds to $(d_✓ : \text{WP}; d_✗ : \text{Wk})$; ✓:Wk,✗:WP corresponds to $(d_✓ : \text{Wk}; d_✗ : \text{WP})$. The blocks highlighted in blue represent the pairs when there is a significant difference ($p$-value $< 0.01$) between the model's answer between $(d_✓ : \text{WP}; d_✗ : \text{Wk})$ and $(d_✓ : \text{Wk}; d_✗ : \text{WP})$ based on McNemar's test.

is shown in Table 7 in the Appendix.

We consider two ways to incorporate the document source into the prompt: (1) *URL*: we add the following sentence for each document: "Webpage $i$ is from the URL: [url]". For each document, we use ChatGPT to extract a keyword from its title, and we use the keyword to construct a URL by concatenating the keyword after pre-defined URL prefixes of each source. For example, the prefix for Wikipedia is `https://en.wikipedia.org/`. Other URL prefixes are shown in Appendix A.2.2. (2) *Name*: We directly tell the LLM the source webpage by "Webpage $i$ is from [webpage name]", where the [webpage name] will be replaced by Wikipedia or WordPress.

## 4.2 Experiment Results

We show the results of CONFLICTINGQA-FAKE in Table 3; the results of CONFLICTINGQA is shown in Table 9 in the Appendix. We have the following observations on the two datasets.

**No% for most models are not higher when the no-document is from a more reliable source.** We only find three models (GPT-4, Tulu, and Llama-3-8B) with No% that are significantly higher when the no-document is from Wikipedia under the direct answer setting. Moreover, Claude-3 sonnet shows a higher No% when the no-document is from WordPress compared with the case when no-document is from Wikipedia.

**Most LLMs mention the sources of the webpages in CoT.** We calculate the proportion of the LLM's CoT answers which contain the source webpage names (Wikipedia or WordPress) to see if the LLM's consider the webpage's source in their

answer. We find that all models, except Llama-2-7B and Llama-3-70B, tend to mention where the documents are from. Regardless of whether the source information is provided by the URL or the name, most LLMs can include this information in their responses. This is a desirable behavior since humans use the source to determine whether a webpage is trustworthy (Tandoc Jr, 2019), so when LLMs retrieve contents from the web to answer the questions, it would be better to include the source webpage's information in their answer for humans to determine whether to trust the answer.

**Changing the documents' source does not have a causal effect on most LLM's answers.** Based on McNemar's test, we find that most LLMs do not change their answer when the sources of the webpages are exchanged when they are prompted to answer directly; for the CoT setting, exchanging the sources has no effect on the LLMs' answers. In summary, while most LLMs know the source differences between the two documents, changing the source does not cause them to change their answer. We also directly query the LLMs "which website is more trustworthy, Wikipedia or WordPress", and LLMs know that Wikipedia is more reliable. However, even though they know Wikipedia is a more reliable, they still do not align their answer with a document from Wikipedia.

## 5 How the Webpages Look

Last, we study whether the webpage's appearance and formatting change the LLM's answer. Since LLMs only use texts as the input in RAG, we are unaware of prior works that consider how the webpage appearance affect the LLM's answer in RAG.

| LLM | Direct Answer | | | | | | CoT | | | | | |
|---|---|---|---|---|---|---|---|---|---|---|---|---|
| | *Screenshot* | | | *Screenshot+Text* | | | *Screenshot* | | | *Screenshot+Text* | | |
| | No% | | Flip ratio | No% | | Flip ratio | No% | | Flip ratio | No% | | Flip ratio |
| | ✓:raw ✗:CSS | ✓:CSS ✗:raw | | ✓:raw ✗:CSS | ✓:CSS ✗:raw | | ✓:raw ✗:CSS | ✓:CSS ✗:raw | | ✓:raw ✗:CSS | ✓:CSS ✗:raw | |
| GPT-4o | 94.4 | 97.6 | 4.8 | 99.2 | 98.4 | 0.8 | 16.0 | 23.2 | 36.8 | 17.6 | 18.4 | 27.2 |
| haiku | 79.2 | 10.4 | 85.6 | 80.0 | 53.6 | 46.4 | 59.2 | 5.6 | 90.4 | 46.4 | 28.8 | 65.6 |
| sonnet | 96.8 | 66.4 | 32.0 | 91.2 | 88.0 | 15.2 | 35.2 | 11.2 | 61.6 | 2.4 | 0.8 | 52.0 |
| opus | 68.8 | 26.4 | 56.8 | 64.0 | 56.8 | 55.2 | 33.6 | 7.2 | 72.0 | 0.0 | 1.6 | 36.8 |

Table 4: The No% and the flip ratio (columns in red) on CONFLICTINGQA-FAKE when changing the webpages' sources. ✓:raw,✗:CSS corresponds to ($d_✓$ : raw; $d_✗$ : CSS); ✓:CSS,✗:raw corresponds to ($d_✓$ : CSS; $d_✗$ : raw). The blocks highlighted in blue represent the pairs when there is a significant difference ($p$-value $< 0.01$) between the model's answer between ($d_✓$ : raw; $d_✗$ : CSS) and ($d_✓$ : CSS; $d_✗$ : raw) based on McNemar's test.

## 5.1 Including Webpage Appearance to Inputs

Given a question and two documents, we create two webpages that are formatted differently for those documents. We use two HTML templates to form webpages: (1) Raw HTML: the webpage only contains the title included in the HTML title tag (<h1>) and the content in the HTML span tag (); an example screenshot is shown in Figure 2 in the Appendix. (2) CSS: the webpage uses an HTML5up TXT template. A webpage contains the title and the content and is formatted with proper CSS attributes. An example screenshot is shown in Figure 3 in the Appendix. We ensure the content's font sizes from the two templates are roughly the same.

To allow the LLM to consider the formatting of the webpages, we consider two different methods: (1) *Screenshot*: We directly replace the "Title," "Text," and "URL" parts in Table 1 with the screenshots of the two webpages; the LLM's input will interleave between texts and the screenshots. The screenshots for the two templates have the same size, and all the textual contents (title and texts) are in the screenshot. This is a realistic setting since users can directly take screenshots of webpages and feed them to the LLM; GPT-4o can also directly use screenshots to reason over the content on macOS. (2) *Screenshot + text*: We feed the LLM the screenshot and the text (title and content). The prompts we use are in Appendix E.

The input to the LLMs can be either ($d_✓$ : raw; $d_✗$ : CSS), where the yes-webpage is formatted using the raw HTML and the no-webpage using the CSS, or ($d_✓$ : CSS; $d_✗$ : raw), where the yes-webpage is formatted using the CSS and the no-webpage using the raw HTML.

## 5.2 Vision LLMs (VLLMs)

We use 4 VLLMs (Radford et al., 2021) here: GPT-4o, Claude-3-haiku, sonnet, and opus. Preliminary experiments confirmed that the above models effectively perform optical character recognition (OCR) on screenshots. We exclude open-source VLLMs since most of them are not trained with multiple image inputs and do not have reasonable performance (Liu et al., 2023; Chen et al., 2023).

## 5.3 Experiment Results

We have the following observations from Table 4.

**Claude-3 tends to agree with no-documents from CSS-formatted webpage screenshots.** When only using the webpage screenshots as the input, No% for ($d_✓$ : raw; $d_✗$ : CSS) is always higher compared with the ($d_✓$ : CSS; $d_✗$ : raw). This observation holds across all three Claude-3 models under direct answer and CoT settings. Contrarily, we do not observe this for GPT-4o.

**No% for ($d_✓$ : raw; $d_✗$ : CSS) and ($d_✓$ : CSS; $d_✗$ : raw) merely differ when the input contains image and texts.** When the input includes not only the webpage screenshots but also the texts in the webpage, No% for most LLMs does not differ regardless of whether the no-document is from a CSS-formatted webpage or not. This may be because the LLM solely relies on the texts and neglects the visual features in the screenshot.

**Changing webpages format has causal effects on the LLM's answers.** By McNemar's test, we find when the input only contains the screenshots, exchanging the appearance of the two webpages from ($d_✓$ : CSS; $d_✗$ : raw) to ($d_✓$ : raw; $d_✗$ : CSS) has a significant causal effect to make all Claude-3 models change their answers to No.

**Many reasons why Claude-3 models tend to agree more on CSS-formatted webpages.** We

scrutinize the CoT responses of Claude-3-haiku when the input only contains ($d_\checkmark$ : raw; $d_\chi$ : CSS) webpage's screenshot to understand why the model tends answer No. We find multiple reasons: (1) Haiku misunderstands the content from the yes-document and believes it supports No. (2) Haiku attributes a sentence to the yes-document while the contents it refers to actually are from the no-document. (3) Haiku hallucinates by changing a sentence from $d_\checkmark$ into a sentence supporting No. Interestingly, we do not find Haiku to mention that the two webpages are formatted differently. While hallucination of VLLMs is an active research topic (Li et al., 2023; Liu et al., 2024a), hallucinating textual contents in screenshots when conflicting evidence is presented is a phenomenon not reported before. We leave it as future works to explore more diverse types of screenshots and how VLLM processes them under conflicting evidence settings.

## 6 Conclusion

We explored how non-textual information in retrieved webpages affects LLM responses amid contradictory evidence. We find that all LLMs we use are sensitive to the webpage's publication time and rely on more up-to-date webpages. We also reveal that when providing the LLM with documents from sources of different credibility, exchanging the source of the two documents barely affects the LLM's answer. Lastly, we show that when LLMs are given only the screenshots of the retrieved webpages, changing the formatting of the webpages has a causal effect on some LLM's answers. Our results highlight an aspect not well-explored in previous RAG literature by showing that certain non-textual information has a causal effect on the model's answer. Whether this is desirable is debatable, but it is essential to first recognize this phenomenon.

## Limitation

We see several limitations in our. First, we only explore three types of non-textual information *indenpendently*, while humans may use other types of non-textual information (Wathen and Burkell, 2002) and make their judgment based on the total effects of all non-textual information. Still, we believe that our choice is well-motivated, and the insights and takeaways of this paper are sufficient to share with the research community.

Next, our experiments only add one type of non-textual information to the document and do not consider the effect of multiple non-textual information together. This is different from how real-world retrieval results can contain diverse types of non-textual information, and exploring how all the non-textual information can affect the LLM's answers is an important future work not addressed in our paper.

Last, while we observe some LLMs are sensitive to the change in non-textual information in the LLMs, and some do not, we do not propose a solution to make the LLM more/less sensitive to the non-textual information. This may be considered a limitation for readers seeking actionable and practical guidelines from our paper. We do not aim to train LLMs that are more/less sensitive to non-textual information as it is unclear what the desired behaviors are for the LLMs. Still, in our preliminary experiments, we explore fine-tuning LLMs to make them more sensitive to metadata in the web pages and find this to be quite successful. We leave the results in Appendix F for interested readers.

## Ethics Statements

We do not see significant ethical concerns in our paper. However, since our results show that sometimes one can change the LLM's answer by manipulating the non-textual information in the retrieved webpages, this might be used to construct adversarial websites to fool the LLMs. For example, if someone wants to generate fake news to make the LLMs believe in it, it can create a well-formatted website published recently. This may successfully trick the LLM into believing the information in it, especially when the LLM does not have too much knowledge about the topic in the fake news. We hope our paper reveals the possible vulnerability to favor certain types of non-textual information in LLMs and draws attention to defending against possible attacks.

Another possible ethics concern is whether we are advocating that LLMs should exhibit human-like cognitive biases. Importantly, this paper does **not** advocate that LLMs should exhibit cognitive biases like humans who prefer more recent documents, nor do we say that non-textual information should or should not change the LLM's answer. We only want to observe the role of non-textual information when LLMs answer questions with conflicting evidence.

# Acknowledgement

We thank the reviewers for their constructive feedback, and we try to incorporate most of them in the paper. Cheng-Han Chiang is supported by a Google PhD Fellowship and a Ph.D. scholarship program by Delta Electronics. We thank the National Center for High-performance Computing (NCHC) of National Applied Research Laboratories (NARLabs) in Taiwan for providing computational and storage resources.

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

# A  Supplementary Materials for Datasets

## A.1  CONFLICTINGQA

We download the dataset from `https://github.com/AlexWan0/rag-convincingness`, which is released under MIT license. Since some questions only have documents that support a single stance, we remove those questions for our experiment. At the end, we obtain 355 questions. For each question, we randomly select one yes-document and one no-document from the evidence related to a question, and we fix those questions in our experiments.

## A.2  CONFLICTINGQA-FAKE

We plan to release this dataset under the MIT license. We provide examples of the question and the supporting documents in Table 5. Below, we elaborate on how we construct this dataset.

### A.2.1  Generating Questions and Evidence

We generate the questions using the topics that used to generate CONFLICTINGQA in Wan et al. (2024). The detailed topics are shown in Table 6. After obtaining the question, we use the following prompt to generate a supporting document.

**Prompt for generating the document**  Here is a question about a non-existing entity.
Question: question
Please generate a paragraph with at least 450 words and less than 550 words to support the answer "stance." Here are some guidelines:

1. Make the paragraph convincing by adding technical details.

2. Do not include any specific dates in the paragraph

| Id | Example |
|---|---|

**Id 1**

**Question** Is the 'DentaHeal 400' capable of regenerating tooth enamel?

**Yes-Doc** **Introducing the DentaHeal 400: A Revolutionary Dental Implant for Enamel Regeneration**

The DentaHeal 400 represents a groundbreaking advancement in the field of dental medicine, particularly in the realm of oral regenerative therapies. Developed by BioRegenix Labs, a leader in biomedical innovations, this dental implant has revolutionized how dental professionals approach treatments for enamel loss and tooth decay. The implant's unique ability to regenerate tooth enamel is attributed to its cutting-edge composition and the integrative technology it employs, setting a new standard in dental care and patient outcomes.

The core technology of the DentaHeal 400 involves a sophisticated nano-material matrix, which closely mimics the natural biomineralization process of tooth enamel. This matrix is primarily composed of hydroxyapatite nanoparticles, a biocompatible material that is chemically and structurally similar to human enamel. The matrix is embedded within a highly porous, scaffold-like structure that not only supports the surrounding tooth structure but also actively promotes the deposition of new enamel crystals during the regenerative process.

Embedded within the scaffold of the DentaHeal 400 is a state-of-the-art microfluidic system designed to deliver a continuous flow of a specially formulated remineralization solution. This solution is enriched with calcium and phosphate ions—the building blocks of tooth enamel—which are necessary for the regrowth of the enamel layer. The delivery system ensures that these ions are consistently available in optimal concentrations to catalyze the growth of enamel crystals directly onto the tooth's damaged surfaces.

Moreover, the implant includes a smart sensor system, developed in collaboration with NanoTech Sensory Solutions, which continuously monitors the pH levels and the biochemical environment in the oral cavity. This real-time data allows the implant to adjust the flow of the remineralization solution automatically, ensuring that the enamel regeneration process is optimized according to individual patient needs and varying oral conditions. This personalized approach not only enhances the effectiveness of the treatment but also significantly reduces the time required for the enamel to regenerate.

Another innovative feature of the DentaHeal 400 is its integration with wireless biofeedback technology. This allows for remote monitoring and adjustments by dental professionals via a secure cloud-based platform, ensuring continuous care and adjustment without frequent visits to the dental office. Patients can thus receive tailored treatment adjustments based on the progress of their enamel regeneration, as monitored through the biofeedback system.

The clinical trials of the DentaHeal 400, conducted in partnership with the Global Dental Health Initiative, have shown remarkable results. Patients who received the implant demonstrated a 95% success rate in enamel thickness restoration to levels comparable to natural healthy teeth within months of treatment, a significant improvement over traditional methods which are unable to regenerate enamel.

In conclusion, the DentaHeal 400 not only restores dental function and aesthetics but also brings a preventative approach to tooth decay and loss, potentially reducing the need for more invasive dental procedures in the future. This implant is not just a treatment but a long-term solution that offers hope to those suffering from dental enamel loss, heralding a new era in personalized dental care.

**No-Doc** **Understanding the Capabilities of DentaHeal 400: Advanced Dental Implant Technology Without Enamel Regeneration**

The DentaHeal 400, a recent innovation in the field of dental implants, has garnered considerable attention for its advanced biocompatible materials and micro-engineering techniques. However, when evaluating its capabilities, particularly in terms of regenerating tooth enamel, a clear understanding of its design and function reveals that it does not facilitate enamel regeneration. The primary purpose of DentaHeal 400 is to replace missing teeth by anchoring artificial roots into the jawbone, utilizing a titanium alloy base known for its strength and compatibility with human bone tissue.

While DentaHeal 400 incorporates cutting-edge technology such as nano-textured surfaces to promote osseointegration (the integration of the implant with the bone), the implant itself does not interact with the biological processes involved in enamel formation. Enamel regeneration, a complex biological process, requires the orchestration of multiple cellular and molecular mechanisms that involve ameloblast cells, which are responsible for enamel secretion. The DentaHeal 400, although sophisticated in design, does not include components or mechanisms that influence or replace the function of ameloblasts, nor does it alter the genetic and cellular conditions necessary for enamel biosynthesis.

To clarify further, tooth enamel, once lost, is notoriously challenging to regenerate due to its unique composition and the fact that ameloblast cells do not persist after the initial formation of enamel during tooth development. Innovations such as the EnamelPro V520, another theoretical product in the field of dental technology, aim to mimic enamel properties using synthetic bioactive materials that adhere to the remaining enamel and dentin. However, unlike DentaHeal 400, the EnamelPro V520 does not replace the tooth structure but rather enhances the existing enamel, addressing issues of erosion and decay superficially rather than through regeneration.

Moreover, the architectural integration of the DentaHeal 400 focuses primarily on longevity and stability within the oral cavity rather than on the regeneration of tooth tissues. Its role is essentially structural, not restorative in the biological sense. Any potential misunderstanding about its capabilities might stem from the general public's growing interest in regenerative medicine and the often ambiguous marketing terminology used in the promotion of dental products. Companies developing dental implants, including those hypothetical manufacturers behind the DentaHeal 400, typically emphasize the durability and biocompatibility of the materials, such as the advanced titanium alloy and the ceramic coatings used, which are praised for their mechanical properties and non-reactive nature.

In conclusion, the DentaHeal 400 represents a significant advancement in dental implant technology, providing a viable solution for tooth replacement. However, it does not possess the capability to regenerate tooth enamel, as its function and design are not oriented towards the biological restoration of enamel but rather towards providing a permanent, durable substitute for missing tooth structures. As research continues in the field of biomaterials and regenerative dentistry, future developments may bridge this capability gap, but as of now, the regeneration of enamel remains beyond the scope of the DentaHeal 400's technological achievements.

**Id 2**

**Question** Does 'GlucoBalance' effectively normalize blood sugar levels in Type 2 diabetes within one week

**Yes-Doc** **GlucoBalance: A Revolutionary Synthetic Hormone Transforming Type 2 Diabetes Management**

GlucoBalance, a groundbreaking synthetic hormone, has emerged as a game-changer in the management of Type 2 diabetes, showing promising results in effectively normalizing blood sugar levels within just one week of administration. Developed by leading endocrinologists at the biotechnological powerhouse PanaceaPharm, GlucoBalance has undergone rigorous trials and has received approval from health regulatory authorities after demonstrating remarkable efficacy and safety profiles.

The mechanism of action for GlucoBalance is intricately designed to mimic the body's natural insulin regulation but with enhanced responsiveness to fluctuating glucose levels. Unlike traditional diabetes treatments that often require time to adjust to the body's needs, GlucoBalance works by directly targeting the insulin receptors on cell surfaces. The hormone facilitates a rapid increase in glucose uptake by cells, thereby decreasing blood glucose to normal levels swiftly and efficiently. Additionally, GlucoBalance incorporates a novel molecular component, Regulatide, which aids in the stabilization of insulin release, preventing the common fluctuations that can occur with other diabetes treatments.

Clinical studies conducted by the Institute of Metabolic Science provided a robust data set showcasing GlucoBalance's capabilities. In a double-blind, placebo-controlled study involving over 500 participants with Type 2 diabetes, those treated with GlucoBalance showed a significant reduction in HbA1c levels compared to those who received a placebo. Most impressively, more than 90% of the participants achieved glycemic control within the first week, an unprecedented statistic in diabetes care.

Safety assessments have also highlighted the hormone's benign nature, recording only minimal side effects, all of which were transient and mild. This safety profile stands in stark contrast to many existing diabetes medications, which can come with substantial risks and long-term health implications.

Moreover, the introduction of GlucoBalance has been supported by robust patient education programs and healthcare provider training modules to ensure optimal usage and monitoring practices. Healthcare professionals have praised the hormone for its ease of integration into existing treatment protocols, providing them with a powerful tool in the battle against the chronic complications associated with poorly managed blood sugar levels.

In summary, GlucoBalance represents a significant advancement in the treatment of Type 2 diabetes. With its ability to quickly normalize blood sugar levels, coupled with a high safety profile and ease of use, GlucoBalance not only offers a new hope for patients looking for rapid and efficient glycemic control but also marks a significant milestone in diabetes care management. Its development is a testament to the possibilities within the realm of synthetic biology and pharmacological innovation, setting a new standard for the treatment of chronic conditions such as diabetes. As ongoing research and real-world usage continue to gather more data, GlucoBalance is poised to become a cornerstone in diabetes treatment regimens globally.

**No-Doc** **Evaluating the Efficacy of 'GlucoBalance' in Normalizing Blood Sugar Levels in Type 2 Diabetes Within One Week**

To address whether 'GlucoBalance', a purported new synthetic hormone, can normalize blood sugar levels in individuals with Type 2 diabetes within one week, we must consider several pivotal aspects of endocrinological therapy and diabetes management. Hormone therapies, particularly those involving glucose regulation, demand rigorous scrutiny before conclusions about efficacy can be drawn.

Firstly, the pathophysiology of Type 2 diabetes involves not only the impaired secretion of insulin by pancreatic beta cells but also significant issues with insulin resistance. This means that peripheral tissues in the body do not respond adequately to insulin, necessitating higher levels for glucose management. GlucoBalance, like any other hormone treatment aimed at glucose control, would therefore need to address both insulin secretion and insulin resistance. Achieving such dual functionality in a single hormone formulation is complex and, based on current scientific understanding and technology, not entirely feasible without combined therapeutic approaches.

Furthermore, the pharmacokinetics and pharmacodynamics of any new synthetic hormone would be critical in determining its rapidity and efficacy in action. For a hormone to effectively normalize blood glucose levels within such a short timeframe as one week, it would require an extraordinarily rapid onset of action and optimal bioavailability. Additionally, hormones typically undergo extensive metabolism and excretion processes, which could attenuate their activity and necessitate more prolonged administration to observe significant clinical benefits.

In the realm of clinical trials and medical research, even the most promising therapies undergo phased studies that assess not only efficacy but also safety profiles. A new agent like GlucoBalance would be subjected to this rigorous testing protocol. Initial studies (Phase I and II) focus on safety, dosing, and early indications of efficacy. Only after these phases can a comprehensive Phase III trial potentially demonstrate definitive efficacy. Each phase can take several months to years, and it is during these periods that any significant results are documented and scrutinized.

Additionally, the development of resistance to synthetic hormones is a well-documented phenomenon. Continuous administration can lead to the downregulation of hormonal receptors, making them less effective over time. This adaptive response by the body can mitigate the initial benefits seen with a new treatment like GlucoBalance.

Moreover, considering other adjunct therapies in diabetes management such as Metformin, SGLT2 inhibitors, and GLP-1 receptor agonists, each works through different mechanisms and takes varying durations to substantially impact blood glucose levels. It is implausible for GlucoBalance alone to achieve what these established therapies accomplish, in combination and over extended treatment periods, within just one week.

In conclusion, while the concept of a synthetic hormone like GlucoBalance that swiftly normalizes blood sugar levels is appealing, current medical research and therapeutic protocols suggest that this is highly unlikely. Diabetes management is complex, necessitating a multifaceted approach and time to achieve stable and lasting glucose control. Therefore, the premise that GlucoBalance can effectively normalize blood sugar levels in Type 2 diabetes within one week does not hold up under scientific scrutiny and practical medical understanding.

Table 5: The first two examples in CONFLICTINGQA-FAKE

```
Publishing, Biodiversity, Religion, Digital Rights, Endangered Species,
Biotechnology, Pomology, Virtual Reality, Numismatics, Wilderness
Exploration, Entomology, Pharmacology, Diabetology, Ornithology,
Lepidopterology, Horticulture, Ethology, Paleoclimatology, Product
Design, Seismology, Climate Change, Sustainability, Stomatology,
Rhetoric, Genomics, Intellectual Property, Gemology, Biomathematics,
Philosophy, Karyology, Biomechanics, Telecommunications, Selenology,
Meteoritics, Demographics, Chronobiology, Malacology, Marine
Conservation, Online Learning, Agribusiness, Sustainable Living,
Ecophysiology, Mammalogy, Herpetology, Politics, Web Design,
Cytogenetics, Neuroscience, Bioacoustics, Veterinary Science,
Informatics, Zoogeography, Organic Farming, Cryptocurrency, Ethnobotany,
Data Privacy, Petrology, Real Estate, Rheumatoid, Serology,
Epistemology, Astronomy, Entrepreneurship, Zymology, Melittology,
Pets, Probabilistics, Holistic Health, Evolution, Ichthyology, Aging,
Trichology, Hematology, Gerontology, Hydrology, Neurology, Metallurgy,
Heuristics, Nematology, Nuclear Energy, Conservation, Botany,
Dermatology, Renewable Energy, Robotics, Spelaeology, Gastroenterology,
Psychobiology, Urology, Creationism, Paleo Diet, Virology, Ergonomics,
Veganism, Volcanology, Folklore, Yoga, Paleopathology, Speculative
Fiction, Xenobiology, Anthropology, Theater, Paleobotany, World
Religions, Pop Culture, Anthropometry, Entertainment, Ancient
Civilizations, Poetry, Comics, Animation, Festivals, Archaeology, Dance,
Radio, Etymology, Sports, Otorhinolaryngology, Mycology, Oncology,
Anthrozoology, Criminology, Television, Paranormal, Philology, Forestry,
Aerospace, Somnology, Broadcasting, Cardiology, Cognitive Science,
Quantum Physics, Phylogenetics, Vulcanology, Epidemiology, Nephrology,
Kinematics, Astronautics, Biophysics, Endocrinology, Kinesiology,
Odontology, Pediatrics, Vaccinology, Semiotics, Thermodynamics,
Constitutional Law, Viniculture, Metaphysics, Lexicology, Astrobiology,
Civil Rights, Plastic Surgery, Typography, Venereology, Networking,
Cryptanalysis, Advertising, Graphic Design, Cloud Computing,
Dacryology, Data Science, Thanatology, Toxicology, Human Geography,
Transportation, Etiquette, Public Transport, Phonetics, Neuropathology,
Multiculturalism, Andragogy, Remote Work, Speleology, Telepathy,
Algorithms, Sociology, Bibliography, Oceanography, Work-Life Balance,
Ethics, Bioethics, Endoscopy, Pedagogy, Cartography, Classical Music,
Paleoethnobotany, Manuscripts, Ufology, Revolutions, Paleozoology
```

Table 6: The 191 topics used to generate the questions. These topics are from Wan et al. (2024).

3. Do not mention that the entity is non-existing. You should make the reader believe that everything in the paragraph is real. Do not include any word like 'hypothetical' that will make the readers question the factuality of the paragraph.

4. You can construct more non-existing entities to make the paragraph sound better.

5. The paragraph you generated does not need to be the central argument or theme of the paragraph. It is enough that the paragraph contains sufficient information to support the answer "stance.

**The prompt to verify the stance of the generated paragraph** Here is a question about a non-existing entity.
Question: question
Here is a relevant paragraph about this non-existsing entity.
Paragraph: paragraph

Using the information in the paragraph, answer the question: "question

Please only answer with "Yes" or "No" without saying anything else. Your response can only contain either "Yes" or "No."

### A.2.2 Constructing URLs

We use the following prompt to generate a title and extract keywords from the titles. When GPT-4 does not extract any keywords, the authors manually extract keywords.

**Prompts for generating the titles** Generate a concise title for the following paragraph from a webpage: paragraph. Please only give me the title without saying anything else like "Sure!" or "Here is ...."

**Keyword extraction prompt** You are given a question. Your job is to extract a list of keywords from the question. For example, the question "Can the 'QuickPrint 3000' print 500 pages per minute?" contains ['QuickPrint 3000'], and the question "Is the 'Giant Forest Skink' considered critically endangered?" contains ['Giant Forest Skink']. Please provide the list of keywords in a python list. For example, ['QuickPrint 3000'] or ['Giant Forest Skink']
Your response should only contain a python list without anything else. That is, your response should be able to use the 'eval' function in python to convert it into a list. You should not start the response by 'python list' or anything else. The first charcter of your response should be '['. Question: {question}

After this, we concatenate the keyword with URL prefixes. The {url_keyword} will be replaced with the keyword extracted in the previous step.

1. Wikipedia: https://en.wikipedia.org/wiki/{url_keyword}

2. WordPress: https://{url_keyword}.wordpress.com/

3. CNN: https://edition.cnn.com/{url_keyword}

4. Natural News: https://www.naturalnews.com/{url_keyword}.html

## B  Supplementary Results of CNN/NaturalNews as Sources

We show the results when using CNN/NaturalNews as the sources in Table 7 for CONFLICTINGQA-FAKE. Compared with the results of Wikipedia and WordPress in Table 3, we do not observe significant differences.

## C  Hyperparameters Used in Generating Responses from LLMs

For all LLMs, we use the following sampling parameters to generate the CoT answer.

- temperature: 1.0

- top p: 0.95

- random seed (for LLMs that support random seed): 42

- maximum number of tokens: 512

We use Huggingface transformers (Wolf et al., 2020) to run all the experiments on a cluster with A6000 and A4000 to run LLMs except 70B models. We use 7 V100 to run the 70B models. We quantize all open-source LLMs into 4-bits in our experiment to speed up inference.

## D  Supplementary Results on CONFLICTINGQA

Here, we show the results on CONFLICTINGQA. We do not include Claude models here due to limited monetary resources.

### D.1  Publication Time

The results are shown in Table 8. We find that the results mostly agree with what we see in CONFLICTINGQA-FAKE. For example, most models are sensitive to the publication date under the direct answer setting, and we observe that changing the document publication dates has a causal effect on some LLM's answers. However, we observe that the gap between swapping the non-textual information is not as large as what we see in CONFLICTINGQA-FAKE. We also find that under the CoT setting, the LLM does not strongly prefer more up-to-date evidence. This is possibly because the LLMs are affected by their own stances when answering the questions in CONFLICTINGQA, which is not a problem for CONFLICTINGQA-FAKE.

### D.2  Source of the Webpage

The results are shown in Table 9.

## E  Prompts for Webpage Appearance

**Prompts for *Screenshot*** Here are the screenshots of two websites:
```
Website 1:
[IMG 1]
Website 2
[IMG 2]
Using these two websites, answer the
following question: "[question]"
Use only the information in the above text
to answer the question. Answer with only
"Yes" or "No". Do not answer with anything
other than "Yes" or "No".
```

**Prompts for *Screenshot + Text*** Here are the screenshots and the texts of two websites:
```
Website 1:
"""
Title: [TITLE 1]
Text: [TEXT 1]
……
[IMG 1]
Website 2
"""
Title: [TITLE 2]
Text: [TEXT 2]
"""
[IMG 2]
Using these two websites, answer the
following question: "[question]"
Use only the information in the above text
to answer the question. Answer with only
"Yes" or "No". Do not answer with anything
other than "Yes" or "No".
```

## F  Fine-tuning LLMs to Make It Sensitive to Metadata

We explain how we Tulu-v2-dpo-7b and 13b models to make them more sensitive to metadata, including the website's source and the publication date. We select this model since we fine-tune the LLM using direct preference optimization (DPO) (Rafailov et al., 2024), which is how those two models are aligned. We use DPO because it is hard to define a supervised ground truth for 'sensitivity to metadata'; instead, we only want LLM learns to use reason with metadata. To do so, we use DPO training.

| LLM | Direct Answer | | | | | | CoT | | | | | |
|---|---|---|---|---|---|---|---|---|---|---|---|---|
| | *URL* | | | *Name* | | | *URL* | | | *Name* | | |
| | No% | | Flip ratio | No% | | Flip ratio | No% | | Flip ratio | No% | | Flip ratio |
| | ✓:Nat ✗:CNN | ✓:CNN ✗:Nat | | ✓:Nat ✗:CNN | ✓:CNN ✗:Nat | | ✓:Nat ✗:CNN | ✓:CNN ✗:Nat | | ✓:Nat ✗:CNN | ✓:CNN ✗:Nat | |
| GPT-4-turbo | 84.0 | 76.8 | 16.8 | 82.4 | 76.8 | 17.6 | 13.6 | 10.4 | 25.6 | 11.2 | 6.4 | 32.0 |
| haiku | 99.2 | 99.2 | 0.8 | 100.0 | 99.2 | 0.8 | 38.4 | 42.4 | 56.0 | 37.6 | 33.6 | 55.2 |
| sonnet | 69.6 | 67.2 | 37.6 | 81.6 | 70.4 | 32.0 | 3.2 | 0.8 | 38.4 | 2.4 | 0.8 | 47.2 |
| Llama-2-7B | 0.0 | 0.0 | 3.2 | 0.0 | 0.0 | 0.0 | 70.4 | 73.6 | 35.2 | 27.2 | 32.0 | 56.8 |
| Llama-2-13B | 99.2 | 99.2 | 0.8 | 99.2 | 99.2 | 0.8 | 45.6 | 37.6 | 35.2 | 40.8 | 41.6 | 41.6 |
| tulu-7B | 48.8 | 40.0 | 52.8 | 39.2 | 33.6 | 55.2 | 24.0 | 20.0 | 53.6 | 28.8 | 25.6 | 52.0 |
| Llama-3-8B | 68.0 | 64.0 | 36.8 | 63.2 | 60.0 | 40.0 | 16.8 | 16.0 | 67.2 | 23.2 | 18.4 | 60.0 |
| Llama-3-70B | 92.0 | 95.2 | 8.0 | 90.4 | 88.0 | 12.0 | 30.4 | 24.8 | 58.4 | 25.6 | 28.0 | 60.8 |

Table 7: The No% and the flip ratio (columns in red) on CONFLICTINGQA-FAKE when changing the webpages' sources. ✓:Nat,✗:CNN corresponds to $(d_✓ : \text{Nat}; d_✗ : \text{CNN})$; ✓:CNN,✗:Nat corresponds to $(d_✓ : \text{CNN}; d_✗ : \text{Nat})$. The blocks highlighted in blue represent the pairs when there is a significant difference ($p$-value $< 0.01$) between the model's answer between $(d_✓ : \text{Nat}; d_✗ : \text{CNN})$ and $(d_✓ : \text{CNN}; d_✗ : \text{Nat})$ based on McNemar's test.

| LLM | Direct Answer | | | | | | CoT | | | | | |
|---|---|---|---|---|---|---|---|---|---|---|---|---|
| | *no-today* | | | *today* | | | *no-today* | | | *today* | | |
| | No% | | Flip ratio | No% | | Flip ratio | No% | | Flip ratio | No% | | Flip ratio |
| | ✓:20 ✗:24 | ✓:24 ✗:20 | | ✓:20 ✗:24 | ✓:24 ✗:20 | | ✓:20 ✗:24 | ✓:24 ✗:20 | | ✓:20 ✗:24 | ✓:24 ✗:20 | |
| GPT-4-turbo | 47.6 | 44.8 | 21.1 | 52.4 | 39.4 | 29.6 | 16.1 | 15.5 | 26.5 | 18.0 | 15.2 | 25.1 |
| Llama-2-7B | 0.3 | 0.0 | 2.5 | 0.3 | 0.0 | 2.0 | 23.9 | 23.7 | 46.2 | 20.8 | 18.6 | 53.2 |
| Llama-2-13b | 80.8 | 80.6 | 8.7 | 88.7 | 81.4 | 12.4 | 26.1 | 22.5 | 46.5 | 23.2 | 22.5 | 38.0 |
| tulu-7B | 15.2 | 15.5 | 25.4 | 16.3 | 16.1 | 32.4 | 15.5 | 14.6 | 35.5 | 13.5 | 14.6 | 41.4 |
| Llama-3-8B | 53.0 | 49.6 | 23.9 | 55.2 | 40.8 | 35.2 | 15.8 | 16.1 | 59.4 | 19.7 | 14.9 | 60.8 |
| Llama-3-70B | 53.0 | 49.3 | 27.3 | 63.4 | 45.9 | 33.2 | 32.4 | 33.1 | 38.7 | 32.4 | 26.1 | 46.5 |

Table 8: The No% and the flip ratio (columns in red) on CONFLICTINGQA when changing the website's publication date. ✓:20,✗:24 corresponds to $(d_✓ : 20; d_✗ : 24)$; ✓:24,✗:20 corresponds to $(d_✓ : 24; d_✗ : 20)$. The blocks highlighted in blue represent the pairs when there is a significant difference ($p$-value $< 0.01$) between the model's answer between $(d_✓ : 20; d_✗ : 24)$ and $(d_✓ : 24; d_✗ : 20)$ based on McNemar's test.

| LLM | Direct Answer | | | | | | CoT | | | | | |
|---|---|---|---|---|---|---|---|---|---|---|---|---|
| | *URL* | | | *Name* | | | *URL* | | | *Name* | | |
| | No% | | Flip ratio | No% | | Flip ratio | No% | | Flip ratio | No% | | Flip ratio |
| | ✓:WP ✗:Wk | ✓:Wk ✗:WP | | ✓:WP ✗:Wk | ✓:Wk ✗:WP | | ✓:WP ✗:Wk | ✓:Wk ✗:WP | | ✓:WP ✗:Wk | ✓:Wk ✗:WP | |
| GPT-4-turbo | 52.1 | 50.7 | 18.3 | 51.4 | 55.6 | 19.7 | 16.0 | 13.1 | 27.9 | 14.8 | 14.5 | 26.5 |
| Llama-2-7b | 0.0 | 0.0 | 0.0 | 0.0 | 0.0 | 0.3 | 17.6 | 20.4 | 39.4 | 19.0 | 18.3 | 40.1 |
| Llama-2-13B | 82.1 | 82.9 | 8.5 | 80.9 | 80.9 | 10.0 | 26.8 | 28.2 | 42.3 | 29.1 | 29.9 | 40.2 |
| tulu-7B | 4.8 | 5.1 | 14.8 | 4.3 | 4.3 | 11.1 | 19.7 | 21.8 | 35.2 | 14.5 | 16.0 | 37.9 |
| Llama-3-8B | 23.6 | 23.6 | 35.9 | 23.9 | 23.4 | 33.3 | 17.1 | 16.5 | 55.8 | 16.5 | 15.7 | 50.1 |
| Llama-3-70B | 53.5 | 54.2 | 26.1 | 52.7 | 54.7 | 25.6 | 26.8 | 26.8 | 43.0 | 27.5 | 26.8 | 43.7 |

Table 9: The No% and the flip ratio (columns in red) on CONFLICTINGQA-FAKE when changing the webpages' sources. ✓:WP,✗:Wk corresponds to $(d_✓ : \text{WP}; d_✗ : \text{Wk})$; ✓:Wk,✗:WP corresponds to $(d_✓ : \text{Wk}; d_✗ : \text{WP})$. No blocks are highlighted in blue since there is no significant difference ($p$-value $< 0.01$) between the model's answer between $(d_✓ : \text{WP}; d_✗ : \text{Wk})$ and $(d_✓ : \text{Wk}; d_✗ : \text{WP})$ based on McNemar's test.

The dataset we use is questions from CONFLICTINGQA-FAKE. We generate two responses from Llama-3-8B; one response is prompted when input documents include metadata (publication time or document sources), and the other is prompted without non-textual information. We use the former as the desired response and the latter as the undesired response. We filter the responses from Llama-3-8B to keep the responses containing publication time or document sources when prompted with non-textual information. The resulting dataset contains 1.27K pairs of responses. We fine-tune the models using DPO for two epochs.

We test the resulting model on the test set of CONFLICTINGQA-FAKE generated in a similar pipeline as described in Section 2.1.2. We find that after fine-tuning using the above dataset with DPO, the models indeed are more sensitive to the metadata and they mention the metadata of the retrieved documents more frequently compared with the models before fine-tuning.

**"Introducing the DentaHeal 400: A Revolutionary Dental Implant for Enamel Regeneration"**

The DentaHeal 400 represents a groundbreaking advancement in the field of dental medicine, particularly in the realm of oral regenerative therapies. Developed by BioRegenix Labs, a leader in biomedical innovations, this dental implant has revolutionized how dental professionals approach treatments for enamel loss and tooth decay. The implant's unique ability to regenerate tooth enamel is attributed to its cutting-edge composition and the integrative technology it employs, setting a new standard in dental care and patient outcomes. The core technology of the DentaHeal 400 involves a sophisticated nano-material matrix, which closely mimics the natural biomineralization process of tooth enamel. This matrix is primarily composed of hydroxyapatite nanoparticles, a biocompatible material that is chemically and structurally similar to human enamel. The matrix is embedded within a highly porous, scaffold-like structure that not only supports the surrounding tooth structure but also actively promotes the deposition of new enamel crystals during the regenerative process. Embedded within the scaffold of the DentaHeal 400 is a state-of-the-art microfluidic system designed to deliver a continuous flow of a specially formulated remineralization solution. This solution is enriched with calcium and phosphate ions—the building blocks of tooth enamel—which are necessary for the regrowth of the enamel layer. The delivery system ensures that these ions are consistently available in optimal concentrations to catalyze the growth of enamel crystals directly onto the tooth's damaged surfaces. Moreover, the implant includes a smart sensor system, developed in collaboration with NanoTech Sensory Solutions, which continuously monitors the pH levels and the biochemical environment in the oral cavity. This real-time data allows the implant to adjust the flow of the remineralization solution automatically, ensuring that the enamel regeneration process is optimized according to individual patient needs and varying oral conditions. This personalized approach not only enhances the effectiveness of the treatment but also significantly reduces the time required for the enamel to regenerate. Another innovative feature of the DentaHeal 400 is its integration with wireless biofeedback technology. This allows for remote monitoring and adjustments by dental professionals via a secure cloud-based platform, ensuring continuous care and adjustment without frequent visits to the dental office. Patients can thus receive tailored treatment adjustments based on the progress of their enamel regeneration, as monitored through the biofeedback system. The clinical trials of the DentaHeal 400, conducted in partnership with the Global Dental Health Initiative, have shown remarkable results. Patients who received the implant demonstrated a 95% success rate in enamel thickness restoration to levels comparable to natural healthy teeth within months of treatment, a significant improvement over traditional methods which are unable to regenerate enamel. In conclusion, the DentaHeal 400 not only restores dental function and aesthetics but also brings a preventative approach to tooth decay and loss, potentially reducing the need for more invasive dental procedures in the future. This implant is not just a treatment but a long-term solution that offers hope to those suffering from dental enamel loss, heralding a new era in personalized dental care.

Figure 2: An example of the `raw HTML` template webpage's screenshot.

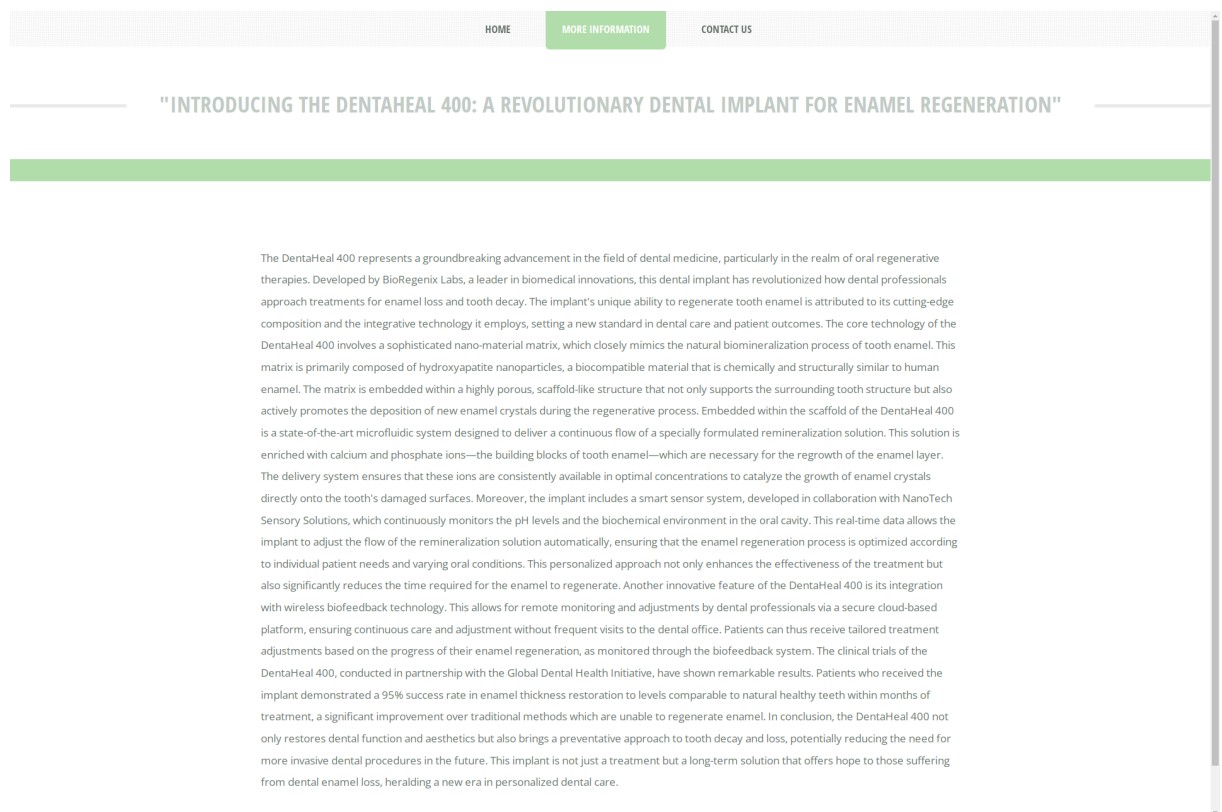

Figure 3: An example of the `CSS` template webpage's screenshot.