# OpenReview forum: "Do Metadata and Appearance of the Retrieved Webpages Affect LLM's Reasoning in Retrieval-Augmented Generation?"
_EMNLP/2024/Workshop/BlackBoxNLP — BlackboxNLP 2024_

### Official Review · Reviewer_9jJM · 2024-09-08

**Overall Assessment:** 4
**Confidence:** 4

**Best Paper:**

1

**Best Paper Justification:**

-

**Comments Questions Suggestions And Typos:**

Row 23-26: Unclear formulation, something is changed in both cases?

Row 280: Missing space between "and" and Y.

Row 396: The statement is true, but the same is true for Tulu (which is not mentioned).

Intro: Using non-textual information to refer to textual remains misleading despite the clarification. Using "metadata" directly would have been more appropriate in my opinion.

Based on the explanation of Section 2.4.2, it remains unclear to me how the proposed proportion comparison would differ from the No% evaluation metric which was later employed. Indeed, looking at the change in negatives when flipping conditions is equivalent to comparing the proportions of positive and negatives, provided that the variable is binary as you state. The connection should be clarified.

**Paper Summary:**

The authors explore whether LLMs that use RAG are sensitive to context perturbations. Specifically, for yes/no questions authors use the ConflictingQA dataset to study whether changing one of three types of metadata (publication date, news source & webpage aesthetics) affects the LLMs’ output. They find that overall LLMs are sensitive to these alterations, with e.g., some models preferring articles that have a more recent publication date. The authors also note that the LLM sensitivity is model-dependent, notably with models of the Claude family being very sensitive to CSS formatting.

**Summary Of Strengths:**

The paper is clearly organized and presents concise but useful findings on the brittleness of LLM to variations in prompt metadata. Several controls were performed to account for influencing factors: many models, real-fake data, direct-CoT query templates, three types of metadata. The authors performed sensible analysis choices to identifying factors with a causal influence in model predictions. Findings are very relevant and can inform future choices in LLM alignment (e.g., paying attention to the credibility/recency of sources when providing answers).

**Summary Of Weaknesses:**

In line 222-226, it is mentioned that the answer can be one of 3 or 4 options. However, in line s 259-260 it is assumed that the LLM answer is binary, which is also used to motivate the usage of McNemar test (line 320). I assume that the answers not falling into the yes/no binary classification were omitted in the evaluation, but this is never mentioned explicitly before the end of Section 2.4.1. In case this is true, additional information should be provided on the proportion of generations not fitting the positive/negative dichotomy, and the binarization should be clearly motivated in light of this result.

The notation employed for the tested settings (the original with $v_1$ assigned to the yes-document and $v_2$ to the no-document, and its flipped variant) is cumbersome and makes understanding unnecessarily complex, especially when combined with causal inference notation. Defining the two settings as e.g. original/flipped would make the explanation clearer and more concise.

Finally, while it is reasonable that comprehensive multi-attribute experiments could not fit the current paper, it would have been very relevant to see how the Publication Time and the Webpage Look setting, which were both found influential for e.g. Claude models in the analysis, would interact when both present (e.g. CSS-styled document that is also less recent, vs. plain one that is more recent). This could have provided reader a contained but useful nugget of information regarding how the properties interact in real-world setups for these models.

---

### Official Review · Reviewer_bwxL · 2024-09-09

**Overall Assessment:** 4
**Confidence:** 4

**Best Paper:**

1

**Best Paper Justification:**

-

**Comments Questions Suggestions And Typos:**

The quality of writing is good and I haven't found any typos or passages that need to be rewritten.

The first paragraph of the limitations section is a bit over the top. It makes more sense to focus the limitations section on specific limitations in your work or specific drawbacks of the technical approach. The first para here essentially says "we could have done more work, but we apologize for just having 8 pages" and adds no information. These points hold for all papers at ACL-style conferences and there is no need to apologize.

**Paper Summary:**

This work investigates the extent to which retrieval-augmented LLMs rely on various types of metadata as well as superficial aspects of the input when deciding what information to base their outputs on. The case that is considered here is when the input includes conflicting information and the LLM has to choose what to rely on. The properties that are investigated are the publication time, the source, and visual style.

The experimental method is based on paired comparisons targeting one type of property: for instance, comparing pairs where one instance is pure text and the other a CSS-formated page. These pairs are created automatically and comparisons are carried out within pairs (using a McNemar test), so the causal conclusions seem sound (since there are no confounders influencing treatment assignment). The experiments are carried out on two datasets: the previously published ConflictingQA dataset, and a new synthetically generated dataset presented here. The purpose of the new dataset is to reduce the confounding effect of LLM training: the LLM may "know" the correct answer which can obscure the effects of the properties under investigation.

The experiments are carried out on a variety of open and commercial models, and it turns out that the models exhibit different behaviors with respect to these properties. (Case in point: the visual style seems to be strongly impactful for the Claude-3 model.)

**Summary Of Strengths:**

- Relevant and timely investigation of input properties that influence the output of RAG models.
- The experimental method is thorough, using a real and a synthetic dataset, and using a paired comparison.
- Interesting effects are seen that differ between models.

**Summary Of Weaknesses:**

- This is solid work and there are no major weaknesses.

---

### Official Review · Reviewer_kJ7g · 2024-09-13

**Overall Assessment:** 2
**Confidence:** 4

**Best Paper:**

1

**Best Paper Justification:**

NA

**Comments Questions Suggestions And Typos:**

NA

**Paper Summary:**

This paper investigates how non-textual information in retrieved webpages affects LLMs' reasoning and answers in RAG settings. The study focuses on three types of non-textual information: publication time, source credibility, and webpage appearance.

**Summary Of Strengths:**

- Novel research question on non-textual information's impact on LLMs in RAG settings.
- Comprehensive experimental design using multiple LLMs and datasets.
- Appreciate that the authors address the ethical implications of their work.

**Summary Of Weaknesses:**

The paper to me does not cover aspects of unravelling the "blackbox" nature of LLMs which is the focus of the conference. While the authors talk about the impact of additional context and type of context on LLMs, the paper does not explore and contribute to improving understanding of how LLMs function.

A couple of other comments regarding the approach and findings:
- The paper aims to understand how LLMs process non-textual information compared to humans, but does not include human performance as a baseline. Including human evaluations on a subset of the data would provide valuable context for interpreting the LLM results.
- While the authors mention in the limitation section that they did not explore the impact of other metadata on the LLMs performance using RAG, they do not just why they chose the three particular attributes that they studies. Did they have any priors for these three attributes having the biggest impact? Also, I think for the sake of the paper it's fine not to include more attributes, but this is something they could have included in appendices. Without the mention of other attributes, the paper feels incomplete.
- The paper examines different types of non-textual information separately. Investigating potential interaction effects (e.g., how publication date and source credibility combine to influence LLM outputs) could yield additional insights.
- The paper could benefit from more discussion on potential strategies to mitigate undesired influences or to leverage beneficial aspects of non-textual information in RAG systems.

---

### Decision · Program_Chairs · 2024-09-19

**Decision:**

Accept

**Comment:**

The paper investigates a relatively neglected question - how meta data affect model's performance in a RAG setting. The experiments are well designed and the results are insightful. One reviewer thought the paper may not fit the workshop because it's not about the model's function, but in fact behavioral analyses are well suited in the workshop.